# Solid-State Fermentation of Plant Feedstuff Mixture Affected the Physiological Responses of European Seabass (*Dicentrarchus labrax*) Reared at Different Temperatures and Subjected to Salinity Oscillation

**DOI:** 10.3390/ani13030393

**Published:** 2023-01-24

**Authors:** Diogo Amaral, Diogo Moreira Filipe, Thais Franco Cavalheri, Lúcia Vieira, Rui Pedro Magalhães, Isabel Belo, Helena Peres, Rodrigo O. de A. Ozório

**Affiliations:** 1Interdisciplinary Centre of Marine and Environmental Research (CIIMAR-UP), 4450-208 Porto, Portugal; 2Department of Biology, Faculty of Sciences, University of Porto, 4169-007 Porto, Portugal; 3Centre of Biological Engineering, University of Minho, 4704-553 Braga, Portugal

**Keywords:** European seabass (*Dicentrarchus labrax*), solid state fermentation, innate immune system, oxidative stress, digestive enzymes

## Abstract

**Simple Summary:**

Climate change is a growing challenge to the aquaculture industry by causing suboptimal farming conditions that reduce fish performance and increase disease outbreaks. Climate change may also limit access to marine and terrestrial animal feed ingredients for aquafeed production, thus creating a need for novel alternatives. Plant feedstuffs in their crude forms have several antinutritional factors that limit their incorporation level in aquaculture feed. However, when these plant ingredients are subjected to fermentation processes, digestibility and bioactive capacity increase, enabling an increase in their inclusion level in the aquafeed formulation. Considering this, we evaluated the effects of a plant feedstuff mixture (PFM: soybean meal, rapeseed meal, sunflower seed, and rice bran) fermented with *Aspergillus niger* on the growth performance and physiological responses of European seabass subjected to environmental stress. Our results demonstrated an interaction effect between the environmental conditions and the novel feed on immune and antioxidant responses. However, the novel feed was unable to improve the performance indicators in seabass subjected to stressful environmental conditions. In fact, the fermented feed caused an inhibition of growth performance, which was not observed in fish fed the non-fermented feed.

**Abstract:**

This study aimed to evaluate the effects of dietary inclusion of plant feedstuff mixture (PFM) pre-treated by solid-state fermentation (SSF) on the physiological responses of European seabass. For that purpose, two diets were formulated to contain: 20% inclusion level of non-fermented plant ingredients mixture (20Mix) and 20Mix fermented by *A. niger* in SSF conditions (20Mix-SSF). Seabass juveniles (initial body weight: 20.9 ± 3.3 g) were fed the experimental diets, reared at two different temperatures (21 and 26 °C) and subjected to weekly salinity oscillations for six weeks. Growth performance, digestive enzyme activities, humoral immune parameters, and oxidative stress indicators were evaluated. A reduction in weight gain, feed intake, and thermal growth coefficient was observed in fish fed the fermented diet (20Mix-SSF). Salinity oscillation led to an increase in weight gain, feed efficiency, daily growth index, and thermal growth coefficient, regardless of dietary treatment. Higher rearing temperatures also increased daily growth index. No dietary effect was observed on digestive enzymes activities, whereas rearing temperature and salinity oscillation modulated digestive enzyme activities. Oxidative stress responses were significantly affected by experimental diets, temperature, and salinity conditions. Catalase and glutathione peroxidase activities showed an interactive effect. Fish reared at 21 °C showed higher enzymatic activity when fed the 20Mix-SSF. Conversely, fish reared at 26 °C showed higher GPx activity when fed the 20Mix diet. Fish reared at 26 °C showed reduced peroxidase and lysozyme activities, while salinity fluctuation led to increased lysozyme activity and decreased ACH50 activity. ACH50 activity increased in fish fed the 20Mix-SSF. Overall, the dietary inclusion of PFM fermented by *A. niger* was unable to mitigate the impact of environmental stress on physiological performance in European seabass. In fact, fermented feed caused an inhibition of growth performances and an alteration of some physiological stress indicators.

## 1. Introduction

Climate change has enhanced the vulnerability of society, ecosystems, and the industrial sector [1]. The changing climate may lead to more frequent and severe extreme weather events, erratic precipitations, and increasing water salinity fluctuations in coastal areas [2]. Aquaculture is an industry heavily dependent and exposed to external environmental parameters, which can lead to many short-term challenges for the sector. In the long-term, climate change can limit access to marine and terrestrial feed ingredients for aquafeed production [3,4].

Water temperature is one of the most influential environmental factors in fish physiology, profoundly affecting the rate of physiological mechanisms [5], feeding behavior and growth performance [6]. Fish have a species-specific optimal body temperature range, ensuring maximum efficiency of physiological processes and overall fitness [7]. Animal performance declines if the environmental temperature rises or falls outside the optimal range. After temperature, environmental salinity significantly affects standard metabolic rate, feed intake, and feed efficiency in fish [8,9]. Suboptimal water salinity during fish development has detrimental effects on growth performance [10], reproduction [11], digestive enzyme activity [12], antioxidant capacity, and immune efficiency [13]. The antagonistic or synergistic effect of exposure to multiple stressors has been little studied, even though in the context of global changes, fish are increasingly exposed to multiple stressors [14]. Indeed, for example, besides being a critical factor in fish physiology, water temperature also plays an interactive role with other abiotic factors on fish performance [15].

In natural habitats, fish respond to environmental changes, seeking more favorable zones to avoid suboptimal conditions [16]. However, in an aquaculture scenario, such behavior is not observed due to the limitation of confinement. These unfavorable environmental conditions undermine fish homeostasis and trigger a stress response, including several biochemical, physiological, and behavioral adaptative responses to the new abiotic factors to restore homeostasis [10,17]. Acute stress leads to transient responses, while chronic or recurrent stress leads to a persistent dysregulation of metabolic activities that may deplete energy reserves, impair physiological fitness, and compromise fish performance [18,19].

Modern aquaculture diets are based on increased levels of plant ingredients used as protein sources, replacing the traditional fisheries-based ingredients. Plant ingredients present several constraints to fish nutrition, including a high level of undigestible carbohydrates, lower protein content, and an unbalanced amino acid profile [20]. These ingredients may undergo biochemical processing, e.g., heat and/or enzyme treatment, or biological processing, such as fermentation, to reduce or eliminate some of the disadvantages [21,22].

Solid-state fermentation (SSF) uses fungi or other microorganisms for the fermentation process of various substrates. Fermentation occurs in the absence (or near absence) of free water and under controlled conditions, similar to the environment to which these microorganisms have adapted to [23], thus enabling higher substrate concentrations [24]. *A. niger*, a filamentous fungus belonging to the *Aspergillus* section nigri, is often selected to be used in SSF due to its easy handling and versatile metabolism [25], being able to ferment a variety of substrates, synthesizing over 19 types of enzymes, such as cellulase, xylanases, ligninases, and pectinase [24,26,27,28], many of them being GRAS (generally recognized as safe) approved by the FDA (Food and drug administration) [29]. The fungal exogenous lignocellulosic enzymes are able to break down the structure of poorly digestible carbohydrates, increasing the digestibility and consequently the bioavailability of nutrients and also the bioavailability of bioactive phenolic compounds, that are otherwise found in insoluble bound forms as conjugates with cellular components [30]. It has also been shown that SSF can increase protein content and decrease anti-nutritional factors, such as phytase [31,32].

Moreover, using fungi in SSF may also enrich the fermented substrate with bioactive molecules, such as lectins, terpenoids, and polysaccharides, with immunostimulatory properties [33]. Fungal polysaccharides, such as β-glucans, are considered potent immunological stimulants, regulating fish growth and immune response, directly activating defense mechanisms such as phagocytic, cytotoxic, and antimicrobial activities, or positively modulating gut microbiota, thus acting as prebiotic substances [34].

European seabass (*Dicentrarchus labrax*) was chosen as the model organism for this study due to its wide tolerance range of temperature (5–28 °C) and salinity (5–50‰) conditions [35,36] with an optimal temperature between 20 and 24 °C [37] and salinity between 33 and 35‰ [38]. In addition, European seabass is the third most produced fish species in Europe and the most important commercially exploited fish in Mediterranean areas [39].

This study aimed to evaluate the effects of dietary incorporation of solid-sate fermented plant feedstuff mixture on the physiological responses of European seabass subjected to osmotic and thermal stress.

## 2. Materials and Methods

### 2.1. Solid-State Fermentation

In a previous study (unpublished data), a plant feedstuff mixture (PFM) composed of soybean meal, rapeseed meal, sunflower seed, and rice bran (25% each) was solid-state fermented with *Aspergillus ibericus* MUM 03.49, obtained from Micoteca of the University of Minho, Braga, Portugal; *A. niger* CECT 2088 or *A. niger* CECT 2915, obtained from CECT (Colección Española de Cultivos Tipo, Valencia, Spain). Solid-state fermentation with the *A. niger* 2088 CECT (Spanish collection of type cultivars) led to high production of carbohydrases and total protein, increased in vitro protein digestibility with a high reduction of fiber content. So, SSF with this fungus species was chosen for diet production. For that purpose, 400 g of PFM was autoclaved (121 °C, 15 min) and inoculated with 80 mL of *A. niger 2088* CECT spore solution (0,1% peptone) at a concentration of 10^6^ spores/mL, the humidity was adjusted to 75% (*w*/*w*) and fermentation was done at 25 °C for 7 days in tray bioreactors (43 × 33 × 7 cm). To ensure oxygenation of the substrate it was mixed every day for the duration of the fermentation.

### 2.2. Experimental Diets

A control diet (42% crude protein, 18% crude lipids), similar to commercial seabass feed, was formulated with practical ingredients containing 20% unfermented PFM. The test diet was formulated similar to the control diet but included 20% fermented PFM (Table 1). Dietary ingredients were ground, thoroughly mixed, and dry-pelleted through a 2 mm die using a pellet mill (California Peter mill, CPM Crawfordsville, IN, USA). Pellets were dried for 48 h at 40 °C, sieved, and stored at −18 °C until usage. Ingredient composition and proximate analyses of the experimental diets are presented in Table 1. Chemical analyses of the diets were performed following AOAC methods [40]

### 2.3. Fish and Experimental Facilities

The fish trial was carried out in the Aquatic Organisms Bioterium (BOGA) facilities of the Interdisciplinary Centre for Marine and Environmental Research (CIIMAR) in Matosinhos (Portugal).

The juveniles of European seabass (*D. labrax*) were acquired from a certified hatchery (SONRIONANSA; Cantabria, Spain). After transport, a total of 162 seabass juveniles (20.9 ± 3.3 g) were quarantined for 15 days in a recirculation aquaculture system (RAS) with a 2000-liter tank and fed a commercial diet (NEO diet, Aquasoja; Sorgal, S.A., Portugal). At the end of the quarantine, fish were transferred to the experimental system and maintained for 7 days for acclimatization purpose. During this period, fish were fed the control diet.

### 2.4. Experimental Design

Fish were distributed into 18 experimental units (9 fish/unit) connected to two identical RAS, in which 12 units were allocated to RAS_1_ and 6 to RAS_2_. In RAS_1_, 6 units were kept at 21 °C and the other 6 units at 26 °C. This was achieved by the implementation of a system of chillers and thermal resistances. Over six weeks, salinity was changed weekly in RAS_1_, simulating the commercial farm conditions in Portugal: 15 → 38 → 21 → 33 → 27 → 45 ppt, having two temperature/salinity groups: 21 °C—Oscillatory salinity and 26 °C—Oscillatory salinity. In RAS_2,_ the 6 experimental units were kept at a fixed temperature (21 °C) and salinity (33 ppt) conditions throughout the trial, acting as the control group (21 °C—Fixed salinity).

For both RAS_1_ and RAS_2_, the water temperature was monitored daily and kept within the selected temperature values: 21.4 ± 0.4 °C (T21) and 25.9 ± 0.3 °C (T26). In RAS_2_, the temperature was constant at 20.9 ± 0.3 °C and the salinity at 33 ± 0.3 ppt. Both RAS had a flow rate at 1.5 L.min^−1^ with continuous aeration and natural photoperiod corresponding to 12 h daylight length. The remaining water parameters were also monitored daily. RAS_1_: oxygen levels (7.6 ± 0.2 mg.L^−1^), pH (7.5 ± 0.2), ammonia (NH_4_^+^) (<0.4 mg.L^−1^), and nitrite (<0.4 mg.L^−1^). RAS2: oxygen levels (7.9 ± 0.1 mg.L^−1^), pH (7.7 ± 0.1), ammonia (NH_4_^+^) (<0.4 mg.L^−1^), and nitrite (<0.4 mg.L^−1^).

During the 6-week trial, fish were hand-fed ad libitum twice a day, each having been experimental variable tested in triplicate.

### 2.5. Sampling

Six fish per treatment were collected and anesthetized with ethylene glycol monophenyl ether (0.25 mL.L^−1^). Weight was recorded to calculate growth performance indicators. Blood samples were collected and centrifuged (10000×*g*; 5 min) to isolate the plasma for the determination of innate immune parameters. The intestine with pyloric caeca was collected for the digestive enzyme activities and the liver for the oxidative stress responses. All samples were stored at −80 °C until analyses.

### 2.6. Growth Performance Parameters

Fish weight was measured individually at the beginning and at the end of the trial. The growth performance indicators were determined as follows: Weight gain (g.ABW kg^-1^.day^-1^) was calculated as 1000 × weight gain (WG) (g)/[(W_i_ + W_f_)/2)]/trial duration (days), where WG (g) was calculated as WG = W_f_ - W_i_, where W_i_ is the initial body weight at the start of the feeding trial and W_f_ is the final body weight. Daily growth index (DGI, % body weight/day) was calculated as DGI = 100 × [(W_f_)^1/3^ - (W_i_)^1/3^]/trial duration (days). Thermal growth coefficient (TGC) was calculated as TGC = 100 × [(W_f_)^1/3^ - (W_i_)^1/3^]/water temperature (°C) × trial duration (days). Feed intake (FI, g.ABW kg^−1^.day^−1^) was calculated as FI = 1000 × feed intake (FI) (g)/[(W_i_ + W_f_)/2)]/trial duration (days), where feed intake (FI, g) was calculated as FI per unit (g)/[(n_i_ + n_f_)/2], where n_i_ is the initial number of fish at the start of the feeding trial and n_f_ is the final number of fish. Feed efficiency (FE, g.g^−1^) was calculated as FE = weight gain (g)/feed intake (FI) (g), and protein efficiency ratio (PER, %) was calculated as PER = (WG (g)/CPI (g)) × 100, where CPI is the crude protein intake, calculate as CPI = FI (g) × (dry feed protein content (%)/100)

### 2.7. Digestive Enzymes Activities

Digestive enzymes activities were measured in the whole intestine with pyloric caeca. Samples were homogenized in 50 mM Tris-HCl and 200 mM sodium chloride buffer (pH 8.0), as described by Rungruangsak-Torrissen [41]. The activity of α-amylase followed the increase in the maltose procedure, by hydrolysis of α-D (1,4) glycosidic bond in polysaccharides, and stained with 3,5-dinitrosalicylic acid as described by Bernfeld [42]. Lipase activity analysis was performed using ρ-nitrophenyl substrate as described by Winkler, et al. [43]. For the proteolytic enzymes, trypsin and chymotrypsin, the assays used were L-BAPNA (N-benzoyl-L-arginine-p-nitroanilide) and SAAPFpNA (succinyl-Ala-Ala-Ala-Pro-Phe-p-nitroanilide) respectively. The formation of nitroaniline was then accounted as a proxy for activity measures [44,45]. Specific enzyme activities were calculated with total protein content, which was analyzed according to the Folin phenol method [46] in the same intestine homogenates.

### 2.8. Oxidative Stress Indicators

Lipid peroxidation (LPO) was assessed by the presence of its by-product, thiobarbituric acid reactive substances [47]. Catalase (CAT) was determined by its action over peroxide hydrogen, according to Claiborne [48]. Glutathione s-transferase (GST) was evaluated by conjugation of glutathione with 1-chloro-2,4-dinitrobenzene [49]. Glutathione peroxidase (GPX) and reductase (GR) were determined by oxidation of reduced nicotinamide adenine dinucleotide phosphate (NADPH) based on Mohandas, et al. [50] and Cribb, et al. [51]. Total glutathione (TG) was evaluated by the formation of 5-thio-2-nitrobenzoic acid, as detailed by Baker, et al. [52]. Oxidized glutathione (GSSG) levels were measured by the concomitant reaction of the reduced glutathione (GSH) with 5,5′ dithiobis-2-nitrobenzoic acid [52]. Finally, the GSH level was calculated as the result of subtracting the amount of GSSG from the TG.

### 2.9. Innate Immune Indicators

Innate immune status was accessed for lysozyme, peroxidase activity and alternative complement pathway (ACH50). Lysozyme is a bacteriolytic protein widely studied on account of its capacity to cleave bacterial peptidoglycans [53]. It was measured by turbidimetric assay based on *Micrococus lysodeikicus* lysis and using hen egg white lysozyme (Sigma, Portugal) as standard [54]. Peroxidase levels were used as an indication of the activation state of circulating leucocytes that are known to increase as a response to infection or stress [55]. Peroxidase levels were determined by 3,3’,5,5’-tetramethylbenzidine hydrochloride (Sigma, Portugal) reduction [56]. Hemolytic activity of the alternative complement system was assayed via rabbit red blood cells (Probiologica Lda., Lisbon, Portugal) agglutination as described by Sunyer, et al. [57], and was defined as the concentration of plasma inducing 50% hemolysis.

### 2.10. Statistical Analyses

Two separate two-way analyses of variance (ANOVA) were performed: Analysis 1: salinity × diet in fish groups reared at 21 °C. Analysis 2: diet × temperature in fish groups subjected to salinity oscillation condition. If the interaction was significant, one-way ANOVA was performed at each temperature or salinity, and a *t*-test was used to assess the effect of temperature or salinity in each dietary treatment. Data transformation was applied when the normality of samples (Grubbs test) was not achieved. Data were checked for homogeneity of variances (Levene’s test). A confidence level of 95% was considered in all statistical analyses. All analyses were performed using the software package IBM SPSS statistics 27—Windows 10.

## 3. Results

### 3.1. Growth Performance

The growth performance and feed utilization efficiency of seabass juveniles held under different environmental conditions are presented in Table 2. Fish kept under oscillatory salinity showed higher FE, DGI, and TGC, regardless of dietary treatment. Fish fed the 20Mix diet showed higher WG than those fed the 20Mix-SSF diet, regardless of the salinity condition. In fish subjected to salinity oscillation, temperature had no effect on TGC, FI, or FE. However, the 20Mix diet led to higher FI and TGC than those that were fed the 20Mix-SSF diet. Fish reared at 26 °C had higher DGI than fish reared at 21 °C.

### 3.2. Digestive Enzyme Activities

In fish subjected to salinity oscillation, α-amylase and lipase activities and α-amylase/trypsin ratio (A/T) were higher, whereas trypsin activity was lower in fish reared at 26 °C when compared to fish reared at 21 °C (Figure 1). Fish fed the 20Mix-SSF diet had a significantly higher trypsin/chymotrypsin ratio (T/C) at 21 °C when compared to fish reared at 26 °C. Within the groups reared at 21 °C, fish subjected to salinity oscillation had significantly lower α-amylase, lipase, trypsin, and chymotrypsin activities than those reared in fixed salinity conditions. In these groups, those fed the 20Mix diet showed significantly higher A/T than those fed the 20Mix-SSF diet.

### 3.3. Oxidative Stress Indicators

Oxidative stress indicators (Figure 2) in fish subjected to salinity oscillation were significantly affected by the rearing temperature and experimental diet. In fish fed the 20Mix diet, GPx activity increased at 26 °C. In fish fed the 20Mix-SSF diet, CAT and GPx activities were significantly higher at 21 °C. Fish fed the 20Mix diet and reared at 21 °C with oscillatory salinity had lower GPx activity than fish fed the 20Mix-SSF diet. The opposite trend was observed in fish reared at 26 °C. Fish reared at 21 °C showed higher GST and GR activities than those reared at 26 °C, regardless of dietary treatment. GSH levels and GSH/GSSG ratio were higher in fish reared at 26 °C than at 21 °C. An interactive effect between temperature and diet was observed in GSSG levels, where fish fed the 20Mix diet and reared at 26 °C had significantly lower GSSG levels than those fed the same diet but reared at 21 °C. Comparing the fish groups reared at 21 °C and subjected to different salinity treatments, a distinct dietary effect was observed, with higher GPX, GR, GSH activities and GSH/GSSG ratio in fish fed the 20Mix-SSF diet, while GSSG levels increased in fish fed the 20Mix diet. GST activity increased, whereas GSH levels and GSH/GSSG ratio decreased in groups subjected to salinity oscillation, regardless of dietary treatment. A significant interactive effect of salinity and diet was observed in TG levels, in which fish fed the 20Mix-SSF diet showed lower TG levels when subjected to salinity oscillation, while no significant effect was observed between salinity groups in fish fed the 20Mix diet. No significant effect of rearing temperature, salinity oscillation, or dietary treatment was observed on lipid peroxidation (LPO).

### 3.4. Humoral Innate Immune Parameters

Humoral innate immune parameters in fish subjected to salinity oscillation were significantly affected by rearing temperature (Figure 3). Peroxidase activity and lysozyme activity decreased at 26 °C, regardless of dietary treatment. Comparing the fish groups reared at 21 °C, lysozyme activity increased, and alternative complement pathway activity (ACH50) decreased in fish subjected to salinity oscillation. In both salinity conditions, fish fed the 20Mix-SSF diet showed higher ACH50 activities than those fed the 20Mix diet.

## 4. Discussion

### 4.1. Growth Performance

Solid-state fermentation (SSF) has been reported to improve the nutritional value of agro-industrial products [31,32], but its utilization for aquafeed production is a novel approach that has been reported to improve overall growth performance in Rohu (*Labeo rohita*) [58,59], Nile tilapia (*Oreochromis niloticus*) [60], rainbow trout (*Oncorhynchus mykiss*) [61], and European seabass [62]. In this study, a general decrease in voluntary feed intake and growth performances was observed in fish fed the fermented diet, which was significant in those maintained under oscillatory salinity, regardless of the water temperature. These results could be potentially explained by reduced palatability associated with the *A. niger* used to ferment the PFM and inclusion levels. For example, in rainbow trout, the dietary inclusion of 13% of *Bacillus subtilis* fermented mixture of soybean and corn gluten meals did not affect growth, but the inclusion of 22% significantly reduced growth performances, while voluntary feed intake was not affected [63]. In broiler chickens, weight gain and feed efficiency linearly decreased with the increased *A. niger* single-cell protein, while feed intake trended to be reduced [64].

Rearing conditions and feed composition play an important role in the growth performance of fish, as both regulate feed intake, metabolism, and digestibility.

Water temperature greatly influences fish growth performance [65]. In this trial, the DGI positively correlated with rearing temperature, as previously reported for seabass [66,67]. However, TGC was not affected by water temperature nor by salinity, confirming the adequacy of the TGC model to determine the effects of different diets of fish maintained under different environmental conditions [68]. The TGC model is relatively unaffected by differences in fish weight and water temperature within the preferred temperature threshold for the species [69] Claireaux, et al. [37] considered the optimal rearing temperature for European seabass to be between 20 and 24 °C, while Conides, et al. [8] proposed higher temperatures (25 to 28 °C). Although this temperature interval is considered a possible stressor, the improved growth performance could outweigh any adverse effects caused by heat stress. No interaction between diet and temperature was found in growth performance and feed utilization parameters meaning that fish fed booth diets responded similarly at the two tested temperatures.

Interestingly, fish reared at 21 °C and subjected to salinity oscillation showed higher WG, FE, and DGI than those reared at fixed salinity conditions. European seabass optimal rearing salinity appears to be a controversial topic. Person-Le Ruyet, et al. [38], Conides, et al. [8], and Dendrinos, et al. [70] reported that the optimal rearing salinity was valued at 30–35 ppt, however Saillant, et al. [71], Eroldogan, et al. [72] and Johnson, et al. [73] reported that European seabass growth performance improved in the salinity range of 15–20 ppt. These authors argued that lower rearing salinity is closer to the isosmotic point between environmental salinity and blood osmolality of euryhaline fish [74], allowing for the least metabolic energy cost for osmoregulation, which can account for 10 to 50% of the total energy budget [10]. Fish reared at 21 °C and subjected to salinity oscillation were reared at six different salinities (15, 38, 21, 33, 27, 45 ppt) throughout the trial. Possibly, the oscillatory salinity group benefited from the lower rearing salinities (15, 21, and 27 ppt) over 3 weeks out of the 6-week trial, allowing for a reduced metabolic energy expenditure on osmoregulatory processes. Indeed, it was demonstrated that hyperosmotic condition (50 ppm) in European seabass, demanded more energy than the hypo-osmotic condition [75]. This metabolic energy sparing effect may have outweighed any stress response to salinity change, resulting in an overall improvement of feed efficiency and growth performance. Accordingly, salinity fluctuation in fish and shrimp led to higher growth rates than constant salinity [76,77,78]. No interactive effect was found between salinity and diet in any measured growth performance and feed utilization parameters meaning that fish fed both diets had a similar response in the two tested rearing conditions.

### 4.2. Digestive Enzymes

The combined effect of temperature and salinity fluctuation on the activity of digestive enzymes in fish has not been studied in detail in euryhaline fish. In the present study, no significant effect of diet was observed on digestive enzyme activities. Similarly, dietary supplementation with a commercial solid-state fermentation product did not affect the trypsin and chymotrypsin activities in rainbow trout (*O. mykiss*) [61] Moreover, the dietary fortification with an extract obtained from the SSF of brewer’s spent grain did not affect the endogenous digestive enzyme activity in European seabass but enhanced feed and protein utilization [79].

In this study, amylolytic and lipolytic enzyme activities increased in fish reared at higher temperatures, as previously reported in other studies [66,80,81], while the opposite was true for trypsin activity. It has already been shown that trypsin activity correlates positively with temperature, but only within the optimal range of the species, so any increase in water temperature above the optimum may lead to a reduction in enzyme activity. Indeed, Ahmad, et al. [82] showed that walking catfish (*Clarias batrachus*) reared at water temperatures between 10 and 35 °C showed higher trypsin and chymotrypsin activity at 25 °C. However, in this trial, no significant effect of temperature was observed on chymotrypsin activity, which, in seabass, has been reported to increase in temperatures below the species optimum range [66].

The protein-sparing effect of dietary lipids is well documented in several fish species [83], while the protein-sparing effect of carbohydrates in European seabass is controversial [84,85,86]. In the present study, despite the higher A/T ratio in fish reared at 26 °C than at 21 °C, it is difficult to conclude whether environmental conditions may enable protein sparing, as no significant effect on the protein efficiency ratio was observed. In groups reared at 21 °C, the higher A/T ratio of fish fed the 20Mix diet correlated well with the higher starch digestibility observed in seabass fed a diet supplemented with an extract from the SSF of brewer’s spent grain [79]. Furthermore, the amylase–trypsin ratio (A/T) was lower than 0.1, which is to be expected in a carnivorous species [87] such as European seabass.

The ratio between trypsin and chymotrypsin activities (T/C) has been interpreted as an indicator of stimulation of growth performance and satiation status [88]. In this study, fish fed the fermented diet and reared at 26 °C had significantly lower T/C than fish reared at 21 °C, which may further indicate that the 20Mix-SSF diet may impair the satiation status, especially in fish reared at higher temperatures.

Salinity fluctuation reduced the overall digestive enzyme activity. This effect has been previously reported in yellowtail kingfish (*Seriola aureovittata*) [89] and gilthead sea bream (*Sparus aurata*) [12]. The reduction in enzymatic activity may be explained by the different water salinities, altering water drinking rates, the pH and ion concentrations, and/or composition of the gut physicochemical properties, possibly affecting the activation of enzyme zymogens and gut transit times [12,90].

### 4.3. Oxidative Stress: Enzymatic and Non-Enzymatic Parameters

During environmental stress conditions, reactive oxygen species (ROS) are generated, potentially causing molecular damage such as lipid peroxidation, protein denaturation, and DNA hydroxylation [91]. To avoid these adverse effects of unbalanced redox status, endogenous antioxidant mechanisms are activated [92]. It has been widely demonstrated that diet composition plays an important role in fish ability to mitigate the effects of oxidative stress [93]. The addition of compounds derived from SSF has also been shown to be an effective strategy for improving the oxidative status of fish, including European seabass [79,94]. In this study, the fermentation process was optimized for the production of lignocellulosic enzymes, which resulted in a significant reduction of bioactive antioxidant compounds in the fermented diets (data to be presented in another publication). No general effect of diet on oxidative status was observed when comparing different temperatures. However, when comparing fixed or oscillatory salinities, the 20Mix-SSF diet led to higher GSH, GSH/GSSG ratio, GR, and GPx activity, but LPO was not affected.

Although higher antioxidant enzyme activity and the ability to improve apparent cellular redox status appear to be beneficial to fish welfare, no significant dietary effect on lipid peroxidation was observed, which may suggest that the higher antioxidant enzyme activity was compensatory for the significantly lower amount of bioactive antioxidant compounds in the 20Mix-SSF diet.

Previous studies reported that LPO in the liver of fish tends to increase when reared at temperatures that differ from the optimal range for the species [66,95], which in the case of seabass is reported to be between 20 °C and 24 °C [37,92]. In this study, however, the water temperature had no significant impact on LPO. GSH increased significantly in fish reared at 26 °C. Consequently, a higher GSH/GSSG, an indicator of the cellular redox status [96], was also observed with increased rearing temperature. Higher glutathione levels are correlated to improved antioxidant defenses [97], as glutathione plays an important role in lipid peroxide detoxification, reducing peroxides to their corresponding alcohols [98]. When comparing the GSH levels with the LPO results in fish reared at higher temperatures, it seems reasonable to infer that the higher glutathione levels may have prevented significant liver lipid damage in fish reared at 26 °C [97].

Glutathione reductase activity was significantly higher in fish reared at 21 °C, possibly due to the need to regenerate GSH and rebalance cellular redox status [96,99]. Higher oxidation of GSH may be an effect of higher activity of glutathione-dependent antioxidant enzymes such as glutathione S-transferase (GST) observed at 21 °C.

Catalase (CAT) and GPx activities have been reported to increase in fish exposed to temperatures outside their optimal range [100,101,102], and in this study, CAT and GPx were affected by temperature and dietary treatment. In fish fed the control diet (20Mix), GPx activity followed the previously reported trend, and CAT activity showed no significant effect of rearing temperature. Interestingly, fish fed the fermented diet (20Mix-SSF) exhibited CAT and GPx activities significantly lower than when exposed to higher temperatures. Indeed, the combined effect of temperature and salinity stress has been reported to reduce CAT and GPx activity [103]. However, this effect was only observed in fish fed the fermented diet, indicating that 20Mix-SSF feed may have offered a lower protective effect to seabass. Nonetheless, since the dietary treatment had no significant effect on LPO, it can be assumed that the observed reduction in CAT and GPx activities did not compromise overall oxidative status.

Environmental salinity has also been observed to affect the oxidative status of euryhaline fish [104]. In the current study, comparing fixed and oscillatory salinity groups (21 °C O.S. and 21 °C F.S.), it was observed that weekly salinity variation resulted in lower TG, mainly due to a reduction in GSH, resulting in reduced GSH/GSSG. The decrease in GSH may be caused by glutathione-dependent GST, whose activity has been widely described in response to osmotic stress [105,106,107]. Indeed, GST activity was 56% higher in fish exposed to salinity fluctuation than in those reared in fixed salinity. This may have contributed significantly to the fact that osmotic stress did not affect hepatic lipid damage in seabass, as the activities of CAT and GPx showed no response to salinity, as also observed by Chang, et al. [104].

### 4.4. Humoral Innate Immunity

Rearing conditions such as water temperature and salinity, as well as the supplementation of immune stimulants, have been shown to influence innate immune responses in fish [13,108,109].

Fish fed the 20Mix-SSF diet showed significantly higher ACH50 activity than those fed the 20Mix diet, which can potentially be explained by the inclusion of the β-glucan-rich cell wall of *A. niger*. Previous studies have reported the immune stimulatory effect of β-glucan in the diet of fish [110,111,112,113]. Indeed, supplementing this polysaccharide has been shown to be an effective strategy for enhancing innate immunity in fish, even when included as an integral component of the fungal cell wall. El-Boshy, et al. [114], Pal, et al. [115], and Chang, et al. [116] showed that feeds containing yeast (*S. cerevisiae*) cell wall preparations and the processed mycelia of mushrooms (*Ganoderma lucidum* and *Coriolus versicolor*) can improve the activity of innate immune system parameters. Chang, et al. [116] observed that orange-spotted grouper (*Epinephelus coioides*) fed 1 g and 2 g of fungal β- glucan per kg of diet had significantly higher ACH50 activity against bacterial infections. The results of this study are in agreement with Bagni, et al. [108], who observed an increase in ACH50 activity in European seabass fed for 30 days on a diet containing β- glucan. Li, et al. [94] reported that turbot (*Scophthalmus maximus*) fed diets supplemented with soybean meal fermented by *Aspergillus awamori* showed higher lysozyme activity. Moreover, Das, et al. [58] reported that Rohu (*L. rohita*) fed diets supplemented with sesame oil cake and mahua oil cake solid-state fermented by *S. cerevisiae* showed higher lysozyme and myeloperoxidase activities. In this trial, however, no significant dietary effect was observed on lysozyme and peroxidase activities. Macrophages, eosinophils, and neutrophils are central cells of the innate immune system, important phagocytic cells, and producers of reactive oxygen species, such as hydrogen peroxide, myeloperoxidase, and peroxidase. Studies on the effects of suboptimal rearing conditions on macrophages and neutrophils have generally reported either no change or environmental modulation of their activity. In this study, peroxidase and lysozyme activities were negatively correlated with temperature, being higher at 21 °C than at 26 °C. This is not consistent with previous studies on the effects of water temperature on peroxidase activity in European seabass [54,66], where rearing temperature had no effect whatsoever. Lysozyme activity has been shown to be seasonal in European seabass [54,117], where reduced activity is observed in response to cold stress. Similarly, Dawood, et al. [103] reported that common carp (*Cyprinus carpio*) subjected to combined osmotic and heat stress showed significantly reduced lysozyme activity and respiratory burst activity than when subjected to isolated osmotic stress. In this study, however, salinity fluctuation led to an increase in lysozyme activity, which was previously reported in Yellowfin seabream (*Acanthopagrus latus*), Asian seabass (*Lates calcarifer*) [13], and rainbow trout (*O. mykiss*) [118]. Mozanzadeh, et al. [13] and Cuesta, et al. [119] reported that Asian seabass (*Lates calcarifer*) and gilthead seabream (*S. aurata*) exposed to osmotic stress by hyposaline and hypersaline conditions, respectively, showed a reduction in complement pathway activity (ACH50). This is consistent with the results observed in the current study, in which European seabass juveniles exposed to salinity fluctuation exhibited reduced ACH50 activity. Such reduction may also be further explained by the fact that these groups of seabass juveniles were exposed to hypersaline conditions (45 ppt) over the last seven days of the environmental stress trial.

## 5. Conclusions

The reduced voluntary feed intake in seabass fed the diet which included 20% of *A. niger* fermented PFM, suggests that diet palatability may be compromised, leading to lower growth performance. Thermal and osmotic stress modulated the physiological response of seabass. Dietary treatment did not influence the digestive enzyme activities and the overall oxidative status but modulated the activities of antioxidant mechanisms. Together, these results may indicate that the reduced antioxidant bioactive compounds in the fermented diet caused a compensatory response by endogenous antioxidant mechanisms. In contrast, fish fed the fermented diet showed enhanced ACH50 activity. The utilization of fermented feedstuffs in aquafeeds under the climate change framework is still a challenge and requires further studies.

## Figures and Tables

**Figure 1 animals-13-00393-f001:**
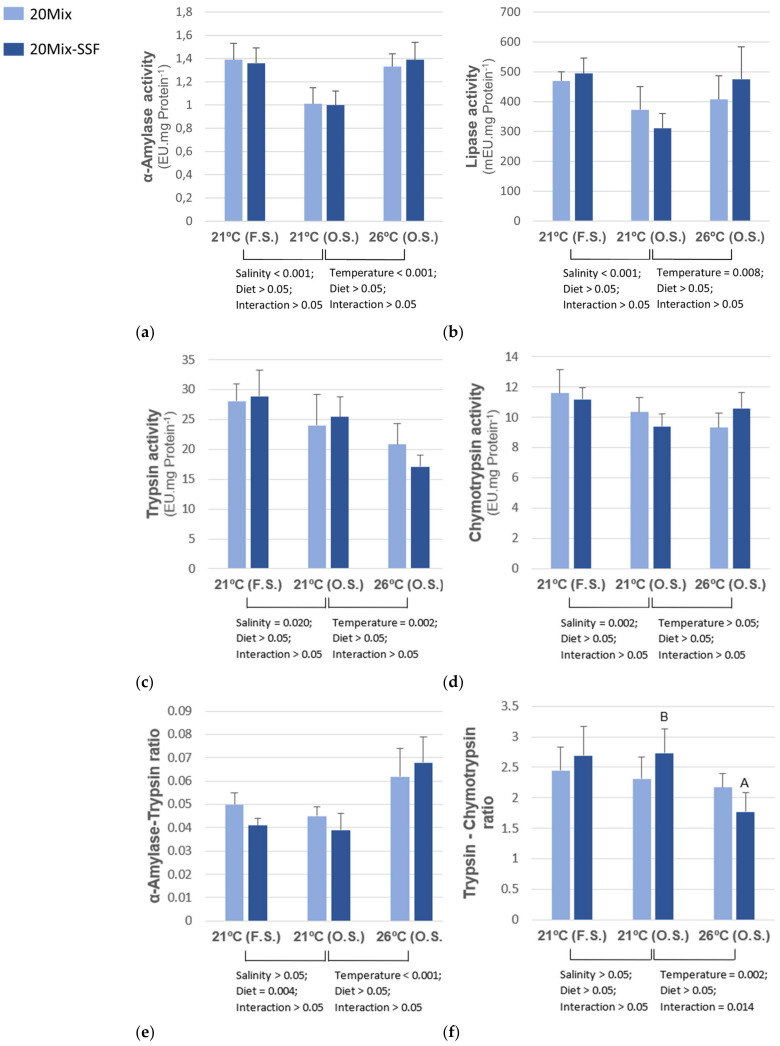
α-Amylase (**a**), lipase (**b**), trypsin (**c**), chymotrypsin (**d**), α-amylase to trypsin ratio (**e**) and trypsin to chymotrypsin ratio (**f**) in *D. labrax* juveniles, fed two experimental diets (20Mix and 20Mix-SSF) and reared at different temperatures (21 °C and 26 °C) under oscillatory salinity (S.O.) and in groups reared at 21 °C subjected to oscillatory (O.S.) or fixed salinity (F.S.). Data (means ±  SD) analyzed using two-way ANOVA. Two-way ANOVA 1: Salinity × Diet, comparing groups reared at 21 °C and subjected to oscillatory salinity and fixed salinity. Two-way ANOVA 2: Temperature × Diet, in groups reared at 21 °C and 26 °C and subjected to oscillatory salinity. If the interaction was significant, one-way ANOVA was performed at each temperature or salinity and a t-test was used to assess the effect of temperature or salinity in each dietary treatment. Uppercase letters stand for significant differences between temperature groups for each experimental diet.

**Figure 2 animals-13-00393-f002:**
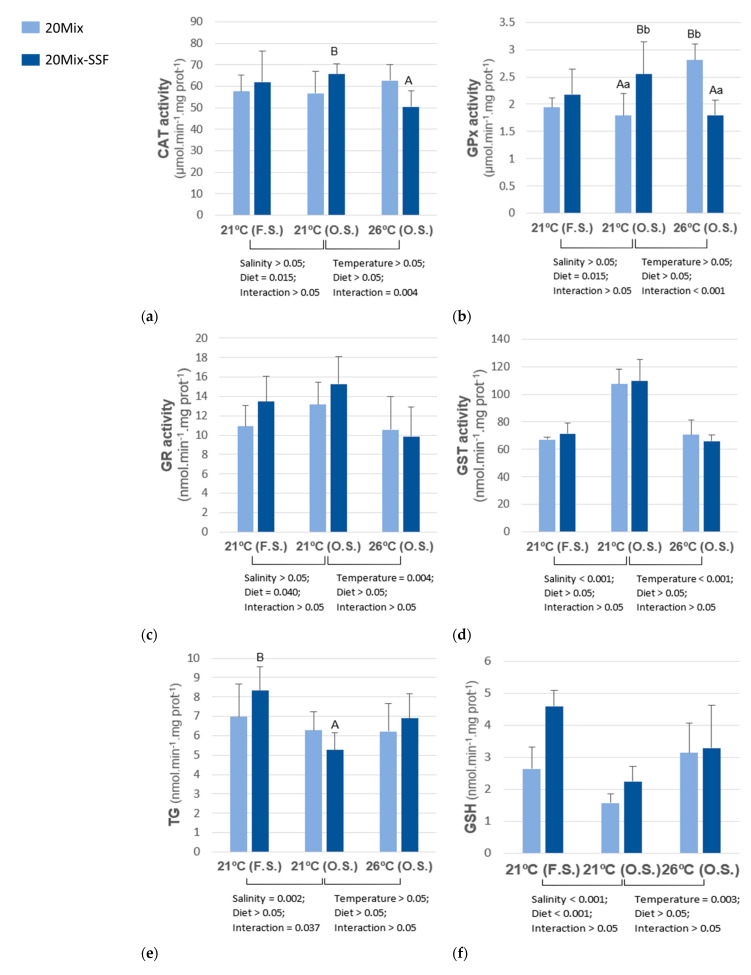
Catalase (**a**), glutathione peroxidase (**b**), glutathione reductase (**c**), glutathione S-transferase (**d**), total glutathione (**e**) reduced glutathione (**f**), oxidized glutathione (**g**), reduced glutathione to oxidized glutathione ratio (**h**) and lipid peroxidation (**i**) in liver of *D. labrax* juveniles, fed two experimental diets (20Mix and 20Mix-SSF) and reared at different temperatures (21 °C and 26 °C) under oscillatory salinity (S.O.) and in groups reared at 21 °C subjected to oscillatory (O.S.) or fixed salinity (F.S.). Data (means  ±  SD) analyzed using two-way ANOVA. Two-way ANOVA 1: Salinity × Diet, comparing groups reared at 21 °C and subjected to oscillatory salinity and fixed salinity. Two-way ANOVA 2: Temperature × Diet, in groups reared at 21 °C and 26 °C and subjected to oscillatory salinity. If the interaction was significant, one-way ANOVA was performed at each temperature or salinity and a t-test was used to assess the effect of temperature or salinity in each dietary treatment. Lowercase letters stand for significant differences among diets and uppercase letters for significant differences between temperature or salinity groups for each experimental diet.

**Figure 3 animals-13-00393-f003:**
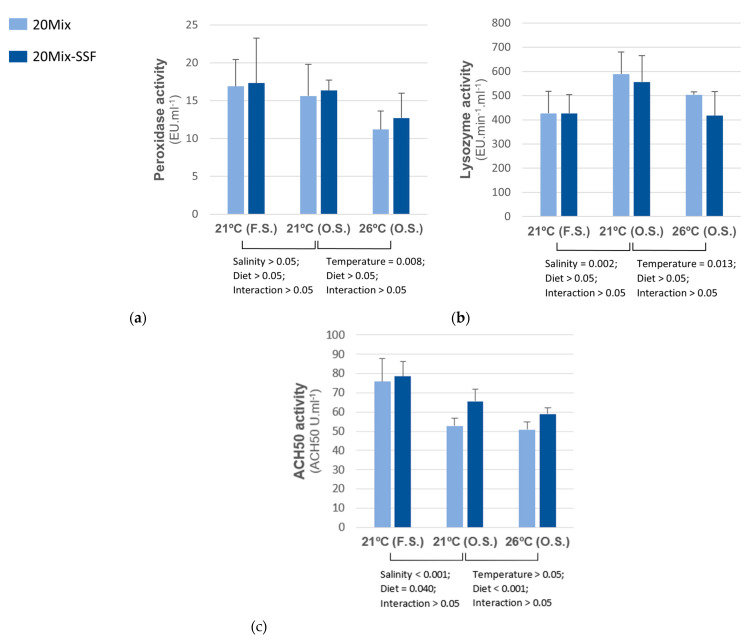
Peroxidase (**a**), Lysozyme (**b**) and alternative complement pathway (**c**) activities in plasma of *D. labrax* juveniles, fed two experimental diets (20Mix and 20Mix-SSF) and reared at different temperatures (21 °C and 26 °C) under oscillatory salinity (S.O.) and in groups reared at 21 °C subjected to oscillatory (O.S.) or fixed salinity (F.S.). Data (means  ±  SD) analyzed using two-way ANOVA. Two-way ANOVA 1: Salinity × Diet, comparing groups reared at 21 °C and subjected to oscillatory salinity and fixed salinity. Two-way ANOVA 2: Temperature × Diet, in groups reared at 21 °C and 26 °C and subjected to oscillatory salinity.

**Table 1 animals-13-00393-t001:** Ingredient composition and proximate analysis of the experimental diets.

Feedstuff (% Dry Weight)	20Mix	20SSF-Mix
Fish meal ^1^	17.5	17.5
Fish protein concentrate ^2^	2.5	2.5
Plant feedstuff mixture ^3^	20.0	0.0
Fermented feedstuff ^4^	0.0	20.0
Wheat gluten meal ^5^	7.5	7.5
Corn gluten meal ^6^	13.4	11.4
Hemoglobin ^7^	5.0	5.0
Wheat meal ^8^	14.1	15.5
Fish oil	13.6	14.2
Hydrolyzed Shrimp ^9^	0.5	0.5
Chromium oxide, 50%	0.5	0.5
Vitamin premix ^10^	1.0	1.0
Choline chloride	0.5	0.5
Minerals premix ^11^	1.0	1.0
Binder ^12^	1.0	1.0
Dicalcium phosphate	1.2	1.2
Lysine ^13^	0.2	0.2
Methionine ^13^	0.2	0.2
Taurine ^13^	0.3	0.3
**Proximate analysis (Dry matter basis)**		
Dry matter (%)	96.8	90.8
Crude protein (%)	42.3	41.4
Crude lipids (%)	17.3	17.4
Ash (%)	8.1	9.3
Gross energy (MJ.kg^−1^)	22.9	22.0
Cellulase (U.g^−1^)	nd	8.1
Xylanase (U.g^−1^)	nd	nd
β-glucosidase (U.g^−1^)	nd	3.8
DPPH (µmol trolox equivalents.g^−1^)	16.6	13.0
Total phenols (mg gallic acid equivalents.g^−1^)	9.4	8.9

^1^ Pesquera Centinela, Stean Dried LT, Chile (Crude Protein (CP): 68.45%; Crude Lipid (CL): 12.56%). Sorgal, S.A. Ovar, Portugal. ^2^ (CP: 80.2%; CL: 15.39%). Sorgal, S.A. Ovar, Portugal. ^3^ Mix (%DM) (CP: 33.0%; CL: 5.5%): 25.0, rapeseed meal (CP: 40.0%; CL: 5.8%); 25.0, soybean meal (CP: 51.9%; CL: 3.7%); 20.5, rice bran (CP: 14.2%; CL: 13.2%); 25.0, sunflower seed (CP: 40.2%; CL: 2.86%). Sorgal, S.A. Ovar, Portugal. ^4^ SSF-Mix (%DM) (CP: 38.03%; CL: 2.5%): 25.0, rapeseed meal (CP: 40.0%; CL: 5.8%); 25.0, soybean meal (CP: 51.9%; CL: 3.7%); 20.5, rice bran (CP: 14.2%; CL: 13.2%); 25.0, sunflower seed (CP: 40.2%; CL: 2.86%). Sorgal, S.A. Ovar, Portugal. ^5^ Wheat gluten (CP: 80.5%; CL: 1.0%). Sorgal, S.A. Ovar, Portugal. ^6^ Corn gluten (CP: 62.0%; CL: 2.8%). Sorgal, S.A. Ovar, Portugal. ^7^ Haemoglobin (CP: 91.5%; CL: 0.4%). Sorgal, S.A. Ovar, Portugal. ^8^ Wheat (CP: 14.33%; CL: 2.09%). Sorgal, S.A. Ovar, Portugal. ^9^ Hydrolyzed Shrimp (CP: 69.8%; CL: 2.1%). Sorgal, S.A. Ovar. ^10^ Vitamin premix (mg kg^−1^ diet): retinol, 18,000 (IU kg^−1^ diet); calciferol, 2000 (IU kg^−1^ diet); alpha tocopherol, 35; menadione sodium bis., 10; thiamin, 15; riboflavin, 25; Ca pantothenate, 50; nicotinic acid, 200; pyridoxine, 5; folic acid, 10; cyanocobalamin, 0.02; biotin, 1.5; ascorbyl monophosphate, 50; inositol, 400. ^11^ Mineral premix (mg kg^−1^ diet): cobalt sulphate, 1.91; copper sulphate, 19.6; iron sulphate, 200; sodium fluoride, 2.21; potassium iodide, 0.78; magnesium oxide, 830; manganese oxide, 26; sodium selenite, 0.66; zinc oxide, 37.5; dicalcium phosphate, 8.02 (g kg^−1^ diet); potassium chloride, 1.15 (g kg^−1^ diet); sodium chloride, 0.4 (g kg^−1^ diet). ^12^ Binder; Aquacube, Agil, UK. ^13^ Feed grade-amino acids. Sorgal, S.A. Ovar, Portugal.

**Table 2 animals-13-00393-t002:** Growth performance and feed utilization efficiency of European seabass juveniles held under different environmental conditions.

Salinity	Fixed		Oscillatory				SEM
Temperature	21 °C		21 °C		26 °C		
Diet	20Mix	20Mix-SSF	20Mix	20Mix-SSF	20Mix	20Mix-SSF	
Initial body weight (g)	20.25	20.89	20.53	20.19	21.09	22.13	1.15
Final body weight (g)	30.52	28.60	35.73	31.61	40.84	38.17	2.39
Weight gain (g.ABW kg^−1^.day^−1^)	9.67	7.46	12.78	10.40	15.32	12.69	1.03
Feed intake (g.ABW kg^−1^.day^−1^)	17.28	15.30	19.51	16.76	20.35	17.58	0.83
Feed efficiency	0.56	0.49	0.66	0.62	0.75	0.73	0.04
Protein efficiency ratio	1.33	1.18	1.55	1.51	1.77	1.76	0.09
Daily growth index	0.95	0.73	1.32	1.04	1.63	1.33	0.12
Thermal growth coefficient	0.45	0.35	0.63	0.49	0.63	0.51	0.04
	**Two-way ANOVA 1**			**Two-way ANOVA 2**			
	Salinity	Diet	Interaction	Temperature	Diet	Interaction	
Final body weight (g)	ns	ns	ns	ns	ns	ns	
Weight gain (g.ABW kg^−1^.day^−1^)	*	*	ns	ns	ns	ns	
Feed intake (g.ABW kg^−1^.day^−1^)	ns	ns	ns	ns	*	ns	
Feed efficiency	*	ns	ns	ns	ns	ns	
Protein efficiency ratio	ns	ns	ns	ns	ns	ns	
Daily growth index	*	ns	ns	*	ns	ns	
Thermal growth coefficient	*	ns	ns	ns	*	ns	

Values presented as mean and pooled standard error of the mean (SEM). ABW: average body weight (initial body weight, IBW + final body weight, FBW)/2. Two-way ANOVA 1: Salinity × Diet, comparing groups reared at 21 °C and subjected to oscillatory salinity and fixed salinity and fed the experimental diets. Two-way ANOVA 2: Temperature × Diet, in groups reared at 21 °C and 26 °C and subjected to oscillatory salinity and fed the experimental diets. Two-way ANOVA: NS—non-significant (*p* ≥ 0.05); * *p* < 0.05.

## Data Availability

The data presented in this study are available on request from the corresponding author.

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
