# Peer review of "Solid-State Fermentation of Plant Feedstuff Mixture Affected the Physiological Responses of European Seabass (Dicentrarchus labrax) Reared at Different Temperatures and Subjected to Salinity Oscillation"

_animals, 2023, doi:10.3390/ani13030393_

Round 1

Reviewer 1 Report

The Authors performed an extensive study of the effects of experimental feeds (containing raw or fermented plant feedstuff mixture), water temperature and two salinity regimes (stable and variable) on growth parameters, activity of digestive enzymes, antioxidant response and nonspecific humoral immune parameters in European seabass. The results showed that fish growth and performance were more dependent on temperature and salinity than on feeding, and that fermentation of plant material did not improve the effects of rearing, probably due to the reduced palatability of fermented feed. 

However, I have some doubts and remarks:

1. Why no control group was used fed plain commercial feed without either nonfermented or fermented plant mixture? And why no 26oC fixed salinity group was used?

2. The chapter 2.4. should contain description of experimental groups with the acronyms used in Table 2

3. Six fish per treatment is quite little (especially that some parameters showed quite high standard deviations)

4. The Discussion is interesting and complete and the Conclusion is supported by the results but I consider this study a preliminary report and suggest to deepen the analysis of the effects of all factors: feeding, temperature and salinity individually. 

Author Response

Point 1: Why was no control group was used fed plain commercial feed without either non-fermented or fermented plain mixture? And why no 26ËšC fixed salinity group was used?

Response 1: Authors thank the reviewer for highlighting such relevant points. In fact, we totally agree with the reviewer, regarding the importance of the inclusion of a commercial diet as a control. In the present trial, it was decided that a commercial diet could not be used as we had no control on the feed formulation, which for the purpose of the current study, is essential. So, in alternative, we decided to formulate the control diet similarly to the actual commercial diet formulation, including low fish meal and high plant feedstuffs levels. This critical information was included in the manuscript (material and method section). Furthermore, a 26ËšC fixed salinity group was not included due to logistical constraints, as no additional experimental units were available in the system.

Point 2: The chapter 2.4. should contain description of experimental groups with the acronyms used in table 2

Response 2: The recommended change was made in the manuscript

Point 3: Six fish per treatment is quite little (especially that some paramaters showed quite high standard deviations)

Response 3: Thank you for the observation. Of course that it would possibly reduce the standard deviation by increasing the number of fish analyzed in each treatment. However, the size sample applied in this study was about 21% of the total fish used in each treatment, which is considered sufficient to obtain reliable biochemical results, since the fish were siblings. Furthermore, there are previous studies in high rank journals as Animals, from different research groups with a similar or inferior number of replicates for digestive enzymes activities [1], oxidative stress markers [2], or plasmatic immune parameters [3,4]

  1. Talukdar, A.; Deo, A.D.; Sahu, N.P.; Sardar, P.; Aklakur, M.; Prakash, S.; Shamna, N.; Kumar, S. Effects of dietary protein on growth performance, nutrient utilization, digestive enzymes and physiological status of grey mullet, Mugil cephalus L. fingerlings reared in inland saline water. Aquacult Nutr 2020, 26, 921-935, doi:https://doi.org/10.1111/anu.13050.
  2. Zhang, J.L.; Zhang, C.N.; Ma, D.D.; Liu, M.; Huang, S.T. Lipid accumulation, oxidative stress and immune-related molecules affected by tributyltin exposure in muscle tissues of rare minnow (Gobiocypris rarus). Fish & Shellfish Immunology 2017, 71, 10-18.
  3. Torrecillas, S.; Roman, L.; Rivero-Ramirez, F.; Caballero, M.J.; Pascual, C.; Robaina, L.; Izquierdo, M.S.; Acosta, F.; Montero, D. Supplementation of arachidonic acid rich oil in European sea bass juveniles (Dicentrarchus labrax) diets: Effects on leucocytes and plasma fatty acid profiles, selected immune parameters and circulating prostaglandins levels. Fish & Shellfish Immunology 2017, 64, 437-445.
  4. Chen, C.Y.; Chen, J.S.; Wang, S.Q.; You, C.H.; Li, Y.Y. Effects of different dietary ratios of linolenic to linoleic acids or docosahexaenoic to eicosapentaenoic acids on the growth and immune indices in grouper, Epinephelus coioides. Aquaculture 2017, 473, 153-160.

Point 4: The discussion is interesting and complete, and the Conclusion is supported by the results but I consider this study a preliminary report and suggest to deepen the analysis of the effect of all factors: feeding, temperature and salinity individually.

Response 4: This study is part of funded large research project in which several trials were carried out in seabass to evaluate the beneficial effects of A. niger against environmental oscillations. We obtained a massive amount of data, that would be unreasonable to present in one publication due to the complexity of the experimental design, we decided then to present the results obtained in different publications. Additional information was added in the conclusions to highlight the need to further studies these aspects.

Reviewer 2 Report

The work presented here describes the influence of two diets (fermented non fermented) and its modulatory function on digestion and immunity under two abiotic stress in European seabass. The design is confusing and unable to draw a solid conclusion as observed. The key focus of a research is missing. Authors try to address several problems altogether, and may be this was one of the reason for such a complex design. My few queries that need to be addressed by the authors are appended below:

1. What is the major issue in farming of this species? whether it is improper plant stuff digestibility or the stress relating to future climatic regimes?

2. A reduction in weight gain, feed intake and thermal growth coefficient was observed in seabass fed 36 20Mix-SSF, with salinity oscillation leading to increased weight gain.....? I think the sentence itself is contradictory.

3. Authors report change in key enzymatic function in different groups. For such cases, an integrated biomarker analysis seems suitable.

4. The conclusion in both abstract and main file is not strong. Conclude with a strong take-home message.

5. The salinity oscillation as performed here is in accordance with the farming condition in Portugal as stated. Whether this range of fluctuation occurs widely, and what is the reason. Because your work should be able to address their problems. 

6. Plant based ingredient fermentation works well for other species, including carnivores. You mentioned the low feed intake relates to low palatibility. Did you check the feeding behaviour?

7. Lastly, the significant supercripts (a,b,c....) are missing in figures. Include.

Kindly address the above referred comments to have clarity of the work.

Author Response

Point 1: What is the major issue in farming of this species? Whether it is improper plant stuff digestibility or the stress relating to future climatic regimes?

Response 1: These issues are not species-specific and have the potential to affect the production of all carnivorous species and fish farmed in open sea cages. Furthermore, it can't really be said that on issue is more urgent or impactful than the other since these issues impact different aspects of aquaculture production, where the focus on developing alternative ingredients to fishmeal, which plant-based ingredients are one of the possible substitutes, regards the economic and environmental sustainability of the sector and the development of functional ingredients with health benefits that may mitigate the impact of environmental or handling stressors impacts the animal wellbeing aspect of the industry. Further research is needed to diversify ingredient options for aquaculture, including nutritional and functional ingredients, to provide greater resilience, as the sustainability of aquaculture is challenged by the scarcity, and high prices of ingredients and the possible climate change impacts on fish physiology and performance.

Point 2: A reduction in weight gain, feed intake and thermal growth coefficient was observed in seabass fed 20Mix-SSF, with salinity oscillation leading to increase weight gain…? I think the sentence itself is contradictory.

Response 2: Authors agree with the reviewer's comments. Indeed, the sentence could be better constructed. Changes were made accordingly.

Point 3: Authors report change in key enzymatic function in different groups. For such cases, an integrated biomarker analysis seems suitable

Response 3: In the present study, a multi-biomarker approach was used, including changes in key enzymatic activity and innate immune indicators to evaluate the effect of the diet and environmental challenges. These biomarkers were chosen based on an extensive collection of biomarkers for aquaculture species done under the framework of the European ARRAINA project “Advanced Research Initiatives for Nutrition & Aquaculture” that identify easy and highly informative nutritionally-regulated biomarkers for the five fish species of the project (Atlantic salmon, rainbow trout, carp, European sea bass, gilthead sea bream).

https://arraina-biomarkers.nutrigroup-iats.org/pages/get/about

Point 4: The conclusion in both abstract and main file is not strong. Conclude with strong take-home message.

Response 4: Authors acknowledge the reviewer's comments. The recommended change was made in the manuscript

Point 5: The salinity oscillation as performed here is in accordance with the farming condition in Portugal as stated. Whether this range of fluctuation occurs widely, and what is the reason. Because your work should be able to address their problems

Response 5: In fact, the salinity oscillation protocol applied in the current study was a result of real salinity data observed in Ilhavo, a Portuguese region with strong aquaculture production and a strong dynamic temperature and salinity oscillations. A warming climate and disrupted water cycles may lead to greater water evaporation and erratic precipitation over the oceans, which can increase the impact of salinity fluctuations on open cage production units. The magnitude and frequency of these fluctuations is also region dependent and different farms may have different experiences. change of environmental salinity is considered a stressor that may undermine the physiological responses of aquaculture fish, and thus functional ingredients with health benefits may mitigate the impact of this stressor in fish physiology.  

Point 6: Plant based ingredient fermentation works well for other species, including carnivores. You mentioned the low feed intake relates to low palatability. Did you check the feeding behavior?

Response 6: The authors believe that the significant problem, in this study, was not the fermentation process, but the species of fungus utilized (Aspergillus niger) which led to unforeseen consequences regarding feed intake, that are not usually observed in other studies. As fish were fed by hand, the feeding behavior was observed in during each meal, and it further contributed to the theory that palatability was the main issue since seabass were observed to actively reject the feed, even at times spitting out the pellets.

Point 7: Lastly, the significant superscripts (a,b,c…) are missing in figures. Include

Response 7: The data presented in the figure showed the effect (two-way ANOVA values) of each independent variable (diet vs temperature or diet vs salinity fluctuation) as well as the interaction between these independent variables. However, when there are only two levels of each independent variable, differences are self-evident, as one is higher than the other. Thus, for readability purposes, we do not consider necessary to add superscripts to the figures to represent the differences detected by the two-way ANOVA. Moreover, if the interaction was significant, a one-way ANOVA was performed at each temperature or salinity, and a t-test was used to assess the effect of temperature or salinity in each dietary treatment, and superscript letters were used to represent the statistical differences. Our group have published previously in such a way. Please see the following publication:

Magalhaes et al. 2022. Aquaculture Research 53:6007–6019 (DOI: 10.1111/are.16070).  

Round 2

Reviewer 2 Report

Dear Authors,

The suggested comments are cautiously taken care. No further comments.